# Pneumothorax after computed tomography-guided lung biopsy: Utility of immediate post-procedure computed tomography and one-hour delayed chest radiography

**Jared Thomas Weinand**[1☯]*, **Lourens du Pisanie**[1☯], **Smith Ngeve**[2☯], **Clayton Commander**[1☯], **Hyeon Yu**[1☯]

1 Division of Vascular and Interventional Radiology, Department of Radiology, The University of North Carolina at Chapel Hill, Chapel Hill, North Carolina, United States of America, 2 The University of North Carolina School of Medicine, Chapel Hill, North Carolina, United States of America

☯ These authors contributed equally to this work.
* jared.weinand@unchealth.unc.edu

**Data Availability Statement:** All relevant data are within the paper and its Supporting Information files.

## Abstract

### Purpose

To evaluate the utility of immediate post-procedure computed tomography (IPP-CT) and routine one-hour chest radiography (1HR-CXR) for detecting and managing pneumothorax in patients undergoing computed tomography (CT)-guided percutaneous lung biopsy.

### Materials and methods

All CT-guided percutaneous lung biopsies performed between May 2014 and August 2021 at a single institution were included. Data from 275 procedures performed on 267 patients (147 men; mean age: 63.5 ± 14.1 years; range 18–91 years) who underwent routine 1HR-CXR were reviewed. Incidences of pneumothorax and procedure-related complications on IPP-CT and 1HR-CXR were recorded. Associated variables, including tract embolization methods, needle diameter/type, access site, lesion size, needle tract distance, and number of biopsy samples obtained were analyzed and compared between groups with and without pneumothorax.

### Results

Post-procedure complications included pneumothorax (30.9%, 85/275) and hemoptysis (0.7%, 2/275). Pneumothorax was detected on IPP-CT and 1HR-CXR in 89.4% (76/85) and 100% (85/85), respectively. A chest tube was placed in 4% (11/275) of the cases. In 3.3% (9/275) of the cases, delayed pneumothorax was detected only on 1HR-CXR, but no patient in this group necessitated chest tube placement. The incidence of pneumothorax was not significantly different between tract embolization methods (p = 0.36), needle diameters (p = 0.36) and types (p = 0.33), access sites (p = 0.07), and lesion sizes (p = 0.88). On logistic

**Funding:** The author(s) received no specific funding for this work.

**Competing interests:** The authors have declared that no competing interests exist.

regression, a lower biopsy sample number (OR = 0.49) was a protective factor, but a longer needle tract distance (OR = 1.16) was a significant risk factor for pneumothorax.

## Conclusion

Following CT-guided percutaneous lung biopsy, pneumothorax detected on IPP-CT strongly indicates persistent pneumothorax on 1HR-CXR and possible chest tube placement. If no pneumothorax is identified on IPP-CT, follow-up 1HR-CXR may be required only for those who develop symptoms of pneumothorax.

## Introduction

Computed tomography (CT)-guided percutaneous lung biopsy is a routinely performed and minimally invasive procedure for a pathologic diagnosis of pulmonary and pleural lesions. While varying as a function of lesion size, CT-guided lung biopsy has a diagnostic sensitivity of close to 90% for lesions measuring at least 1.5 cm in diameter [1]. In a 2011 study on the yield of transthoracic needle biopsy specimens, diagnostic results were obtained in 83% of pleural lesions and 80% of parenchymal lesions [1]. Newer literature suggests more recent improvements, with diagnostic sensitivity reported as high as 97–100% in some cases [2]. As public health initiatives and task force recommendations increase the number of patients undergoing routine lung cancer screening, it is reasonable to expect a commensurate increase in the number of patients with CT-detected lung nodules presenting for diagnostic workups, including CT-guided lung biopsies [3].

Although CT-guided lung biopsies are reported to have a low rate of major complications, they have rarely been associated with potentially serious complications such as hemoptysis, hemothorax, tumor seeding, and air embolization [1, 4]. Pneumothorax has long been established as the most common complication of CT-guided lung biopsy, with rates in the literature historically ranging widely between 8% and 61% [5, 6]. A 2017 meta-analysis suggested overall complication and pneumothorax rates for core-needle lung biopsy were around 38.8% and 25.3%, respectively [4].

There is potential for significant morbidity and mortality from an undetected or delayed pneumothorax. Pneumothorax can be classified as either a major or minor complication based on case-specific factors. In the Society of Interventional Radiology (SIR) Quality Improvement Guidelines for Percutaneous Needle Biopsy, a post-biopsy pneumothorax requiring a chest tube can be designated a minor complication if it only results in a brief overnight hospital stay [5]. However, if the patient with pneumothorax requires more than 48 hours of hospital care, it is deemed a major complication [5].

It has been reported that pneumothorax is most commonly detected immediately after biopsy on final CT images and less commonly identified during recovery (within hours of the conclusion of the procedure), with reported incidences of the latter ranging from 3.3% to 8.6% [6–8]. In Choi et al.'s study in 2004, delayed pneumothorax occurred within 3 hours in 3.3% of the cases, and 20% of those with delayed pneumothorax required chest tube placement [8]. In a more recent study in 2017, Taleb et al. reported an incidence of delayed pneumothorax at 8.6%, with 41.7% of those developing clinically significant pneumothorax [7]. Due to the potential risk of delayed complications, it is generally recommended to observe the patients in a post-procedure care unit to document and monitor recovery from sedation, biopsy-related adverse events, and vital signs [5]. However, post-procedure care strategies vary widely

between institutions, ranging from clinical observation without imaging follow-ups to obtaining a series of chest radiographs after the procedure at various intervals [6, 9, 10].

Therefore, the purpose of this study was to evaluate the utility of immediate post-procedure CT (IPP-CT) and one-hour delayed chest radiography (1HR-CXR) in the detection and management of pneumothorax in patients undergoing CT-guided percutaneous lung biopsy.

## Materials and methods

### Ethics statement

The study was approved by the Institutional Review Board (IRB) in the Office of Human Research Ethics at the University of North Carolina and was compliant with the Health Insurance Portability and Accountability Act. The IRB waived informed consent due to the retrospective nature of the study.

### Study design and patients

All patients who underwent CT-guided percutaneous lung biopsy from May 7, 2014 through August 27, 2021 by the Vascular and Interventional Radiology (VIR) service at a single institution were included. A total of 308 consecutive procedures were identified, and 33 procedures were excluded. Exclusion criteria were 1) patients without 1HR-CXR (n = 11), 2) pediatric patients (age < 18 years, n = 6), 3) ultrasonographic-guided biopsy (n = 9), 4) extra-thoracic mass (n = 3), 5) patients without IPP-CT (n = 1), 6) cases without pathology results (n = 1), and 7) cases aborted due to decreased lesion size (n = 2) (Fig 1). A total of 275 biopsies in 267 adult patients (147 men, 128 women; mean age, 63.5 ± 14.1 years; range 18–91 years) who underwent CT-guided biopsy of lung parenchymal lesions, including pleural-based lesions, with available IPP-CT, 1HR-CXR, and pathologic results were divided into groups with and without pneumothorax and included in the final analysis (Fig 1). Patient demographics and baseline characteristics are summarized in Table 1.

### CT-guided biopsy technique

All procedures were performed on a 64-slice Siemens Somatom CT scanner with CT fluoroscopy capabilities (Siemens Medical Solutions, Erlangen, Germany). Informed consent was obtained and patients were brought to the procedure area. Patients were placed in appropriate position on the CT table based on anatomical location of the target lesion and other patient-specific factors. A combination of intravenous midazolam and fentanyl was used for moderate sedation in all cases where feasible, with a nurse continuously monitoring patient vital signs throughout each procedure. Target lesions were verified and skin entry sites were determined using a grid with radiopaque markers on initial axial CT images. Patients were prepped and draped in sterile fashion. 1% lidocaine was administered at the skin entry site for local anesthesia.

Once the optimal path to the target was chosen to avoid fissures, emphysema, and vessels in the lung and chest wall, a coaxial introducer needle was advanced in the axial plane to the target lesion with intermittent CT images. CT fluoroscopy was commonly used to guide needle placement (n = 187, 68%). Core biopsy specimens were then obtained using various biopsy devices, including CorVocet (Merit Medical Systems, South Jordan, UT), Monopty (BD, Franklin Lakes, NJ), Quick-Core (IZI Medical Products, Owings Mills, MD), and Temno Evolution (Merit Medical Systems, South Jordan, UT). In some cases, the needle tract was embolized using an autologous blood patch or Gelfoam (Pfizer, New York, NY) slurry at the operator's discretion, as detailed in Table 1.

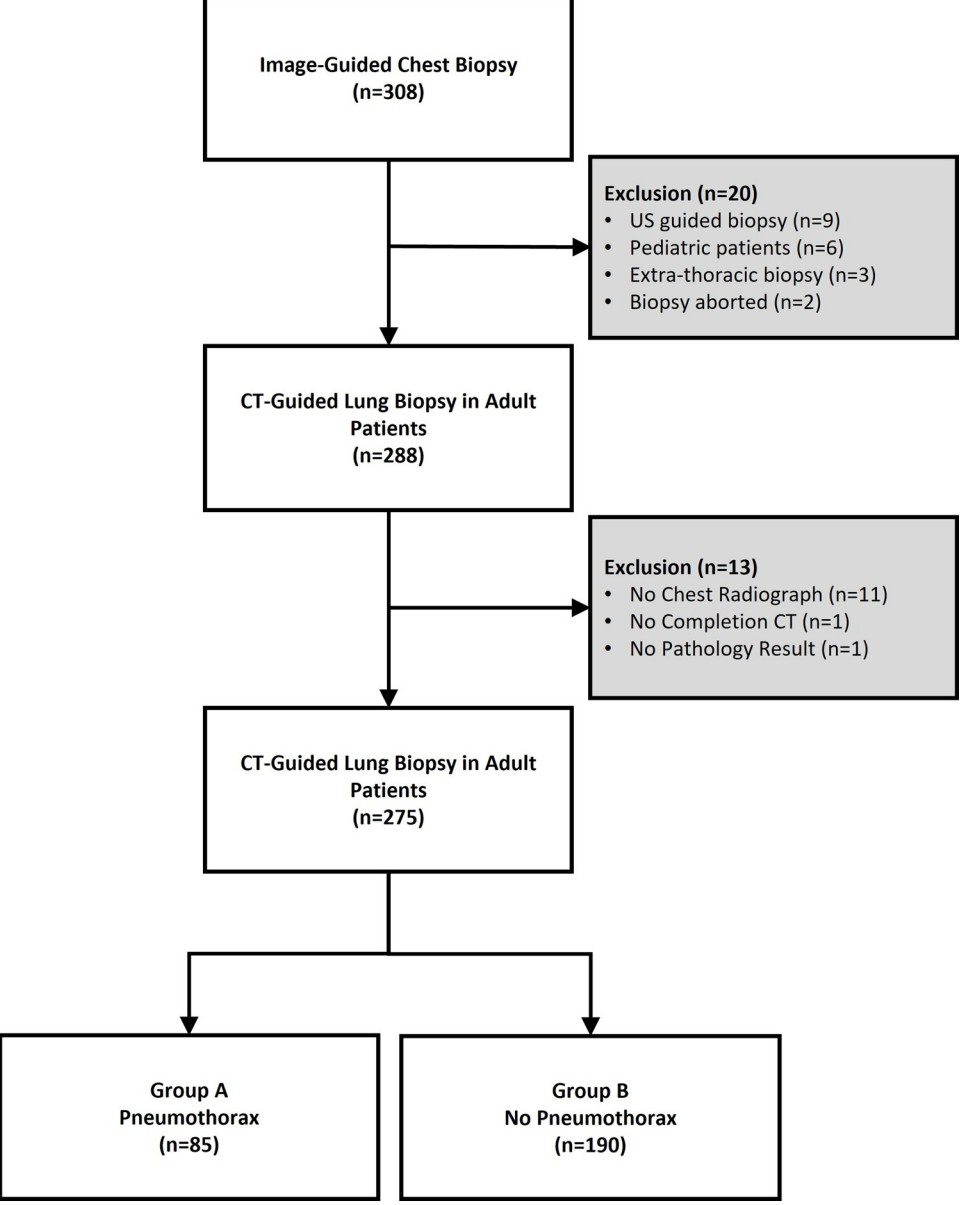

**Fig 1. Inclusion/exclusion criteria flowchart.** 275 CT-guided percutaneous core lung biopsy cases were included and divided into group A (pneumothorax) and group B (without pneumothorax).

Post-procedure CT was performed immediately after removing the coaxial needle in all cases to detect potential complications, including pneumothorax. Portable chest radiography was then performed approximately one hour after the completion of the procedure in all included patients, either at the post-procedure unit for outpatients or in the hospital room for inpatients.

## Pneumothorax management

For a small (distance between the pleura and the lung < 1 cm) pneumothorax confirmed on IPP-CT, aspiration of the pneumothorax with an 18-gauge standard needle or 6-F, 19-gauge

**Table 1. Baseline characteristics.**

| Total number of biopsies | | N = 275 |
|---|---|---|
| Demographics | Age (years) | 63.5 ± 14.1 (18–91) |
| | Female | 128 (46.5%) |
| | Male | 147 (53.5%) |
| COPD[a] | | 35 (12.7%) |
| Lesion size (cm) | | 2.5 ± 2.2 (0.6–16.3) |
| Lesion location | Left upper lobe | 60 (21.8%) |
| | Left lower | 70 (25.5%) |
| | Right upper | 46 (16.7%) |
| | Right middle | 30 (10.9%) |
| | Right lower | 66 (23.9%) |
| | Perihilar | 1 (0.4%) |
| | Posterior mediastinum | 1 (0.4%) |
| | Subpleural | 1 (0.4%) |
| Biopsy device diameter | 20-gauge | 239 (86.9%) |
| | 18-gauge | 36 (13.1%) |
| Biopsy device type | Full-core cutting | 177 (64.3%) |
| | Notched needle | 98 (35.7%) |
| Biopsy device name | CorVocet | 177 (64.4%) |
| | Monopty | 81 (29.4%) |
| | Quick-Core | 16 (5.8%) |
| | Temno Evolution | 1 (0.4%) |
| Needle access | Anterior | 60 (21.8%) |
| | Posterior | 81 (29.5%) |
| | Lateral | 134 (48.7%) |
| Needle access angle (˚) | | 62.3 ± 19.7 (16.7–90) |
| Needle tract distance (cm) | | 1.8 ± 1.8 (0–9.2) |
| Number of samples | | 4.4 ± 2 (2–17) |
| Tract embolization | Blood patch | 227 (82.5%) |
| | Gelfoam slurry | 48 (17.5%) |
| CT fluoroscopy | | 187 (68%) |
| Total DLP[b] (mGy*cm) | | 354.2 ± 3.8.8 (35–3029) |
| Pathology result | Chronic inflammation | 54 (19.6%) |
| | Fungal infection | 3 (1.1%) |
| | Benign tumor | 10 (3.6%) |
| | Malignant: metastasis | 95 (34.5%) |
| | Malignant: primary | 98 (35.7%) |
| | Non-diagnostic | 15 (5.5%) |

Patient/procedure specifics and pathology result data for the 275 biopsy cases.

[a]COPD = chronic obstructive pulmonary disease

[b]DLP = dose length product

Yueh Centesis needle (Cook Medical, Bloomington, IN) was performed. If the size of the pneumothorax did not substantially change or grew, an 8-Fr pigtail chest tube was placed and connected to a Pleur-evac, water-seal chest drainage system (Teleflex, Morrisville, NC). The patients who underwent chest tube placement were continuously monitored and followed up

with serial (one or two-hour) chest radiography until the pneumothorax was resolved and the chest tube was removed.

## One-hour routine follow-up chest radiography

Follow-up 1HR-CXR was formally interpreted and reported by institutional chest radiologists and further reviewed by an interventional radiologist prior to discharge. For patients with small pneumothorax on IPP-CT that had not initially received a chest tube: if the pneumothorax continued to expand or resulted in symptoms, such as chest pain or shortness of breath, an 8-Fr pigtail chest tube was placed with CT or fluoroscopy guidance and the patient was monitored and followed up with serial chest radiography, as described above, until the chest tube was able to be removed. If a small pneumothorax persisted or decreased without symptoms, the patient was continuously monitored for one additional hour without an imaging study before discharge. For patients without pneumothorax on IPP-CT but with pneumothorax seen on 1HR-CXR, the size of the pneumothorax and presence of any other associated symptoms determined whether the patient required chest tube placement.

Per institutional criteria, outpatients were counseled about signs and symptoms that should prompt them to seek urgent medical attention, vital signs were reviewed, and other institutionally defined criteria were met before outpatient discharge in all cases. All medical records were routinely reviewed to evaluate for delayed complications related to biopsy. All outpatients were contacted via telephone approximately 24 hours after discharge for routine post-procedural evaluation by one of the Vascular and Interventional Radiology department nurses. All inpatients were serially monitored with medical record reviews until discharge.

## Statistical analysis

All statistical analysis was performed using R statistical computing language (ver. 4.2.2, R Core Team, 2021) [11]. The Kolmogorov-Smirnov test was used to evaluate the normality of data. For parametric data, the Student-t or Chi-squared test was used to compare variables between groups with and without pneumothorax. For nonparametric data, Mann-Whitney U or Fisher's exact test was used. Continuous variables were expressed as mean ± standard deviation and categorical variables were demonstrated as numbers with percentages. To evaluate the relationship between pneumothorax and associated variables, including: 1) method of needle tract embolization, 2) needle tract distance, 3) needle access site, 4) needle entry angle, 5) biopsy device diameter, 6) biopsy device type, 7) lesion size, 8) lesion location, 9) underlying chronic obstructive lung disease (COPD), and 10) number of biopsy samples, logistic regression analysis was performed using a general linear model and odds ratios were calculated to find the significance of the predictors for pneumothorax. The number of samples taken during a biopsy was divided into a group with sample number $\leq 4$ and a group with sample number $> 4$ to evaluate the relationship of sample numbers with pneumothorax risk. Statistically significant differences were defined as $p < 0.05$.

## Results

### Pneumothorax on IPP-CT

Pneumothorax was identified in 85 cases (30.9%), and most of them (89.4%, 76/85) were detected on IPP-CT. A chest tube was placed in 4% of all cases (11/275), in 12.9% (11/85) of all cases of pneumothorax, and in 14.5% (11/76) of cases of pneumothorax detected on IPP-CT. Chest tubes were removed after confirming resolution of pneumothorax on follow-up chest

radiography the same day (n = 1), the next day (n = 3), in 2 days (n = 3), in 3 days (n = 2), in 4 days (n = 1), and in 7 days (n = 1).

Among all cases of pneumothorax detected on IPP-CT, 51.3% (39/76) completely resolved on 1HR-CXR without needle aspiration or chest tube placement. The size of the pneumothorax (distance between the parietal pleura and the lung) was significantly different between the groups with and without spontaneous resolution (0.5 cm ± 0.4 vs. 0.8 cm ± 0.6, p<0.001) (Fig 2). In 5 patients (6.6%, 5/76), pneumothorax on IPP-CT was aspirated through a needle and recurrence was not seen on 1HR-CXR in any of the cases. All patients with pneumothorax initially detected on IPP-CT but resolved on 1HR-CXR (55.3%, 42/76) were discharged after 1 hour of additional monitoring and no further chest radiography.

## Delayed pneumothorax on 1HR-CXR

There were 9 cases (3.3% of all biopsy cases, 9/275; 10.6% of all pneumothorax cases, 9/85) of delayed pneumothorax (not detected on IPP-CT but identified later on 1HR-CXR). The incidence of delayed pneumothorax among patients who showed no evidence of pneumothorax after a biopsy was 4.5% (9/199). However, none of those patients necessitated chest tube placement. All 9 patients had a small pneumothorax (< 1 cm) on 1HR-CXR without chest pain or shortness of breath. They were discharged the same day after resolution of pneumothorax or no increase in pneumothorax size on follow-up chest radiography. No patients returned to the hospital or emergency department for newly developed symptoms or signs associated with worsening pneumothorax.

## Risk factors for pneumothorax

When compared between groups with and without pneumothorax, there were no significant differences in patient demographics including age (p = 0.69) and gender (p = 0.81), needle tract embolization methods (p = 0.36), needle diameters (p = 0.36) and types (p = 0.33), access sites (p = 0.07), underlying COPD (p = 0.51), and lesion size (p = 0.88), as outlined in Table 2. The average number of biopsy samples was not significantly different between the two groups (p = 0.16). However, when they were divided into a group with greater than 4 samples and a group with 4 or fewer samples, the incidence of pneumothorax was significantly higher in the group with greater than 4 samples (p = 0.01).

Among all associated variables, only the needle tract distance and the number of biopsy samples were significant factors related to pneumothorax. Logistic regression demonstrated needle tract distance to be a significant risk factor for pneumothorax (OR: 1.16; 95% CI: 1.01–1.34) and a smaller number of biopsy samples ($\leq$ 4) to be a protective factor (OR: 0.49; 95% CI: 0.29–0.83) against the development of pneumothorax.

## Diagnostic performance of biopsy

The overall diagnostic sensitivity of the biopsies was 94.6% (260/275), as detailed in Table 1. Even after dividing the lesion sizes into small ($\leq$ 1.5 cm) and large (> 1.5 cm) groups, the diagnostic performances were not significantly different, with diagnostic results in 94.2% (97/103) and 94.8% (163/172), respectively. The diagnostic sensitivities were also similar between the groups with CT fluoroscopy guidance and conventional CT technique. The rates of non-diagnostic samples were 5.3% (10/187) on CT fluoroscopy and 5.7% (5/88) on conventional CT guidance without a statistically significant difference. However, the total dose length product (DLP) was significantly higher in patients with CT fluoroscopy compared with conventional CT (383.8 ± 353.8 mGy*cm vs. 291.3 ± 163.6 mGy*cm, p = 0.003).

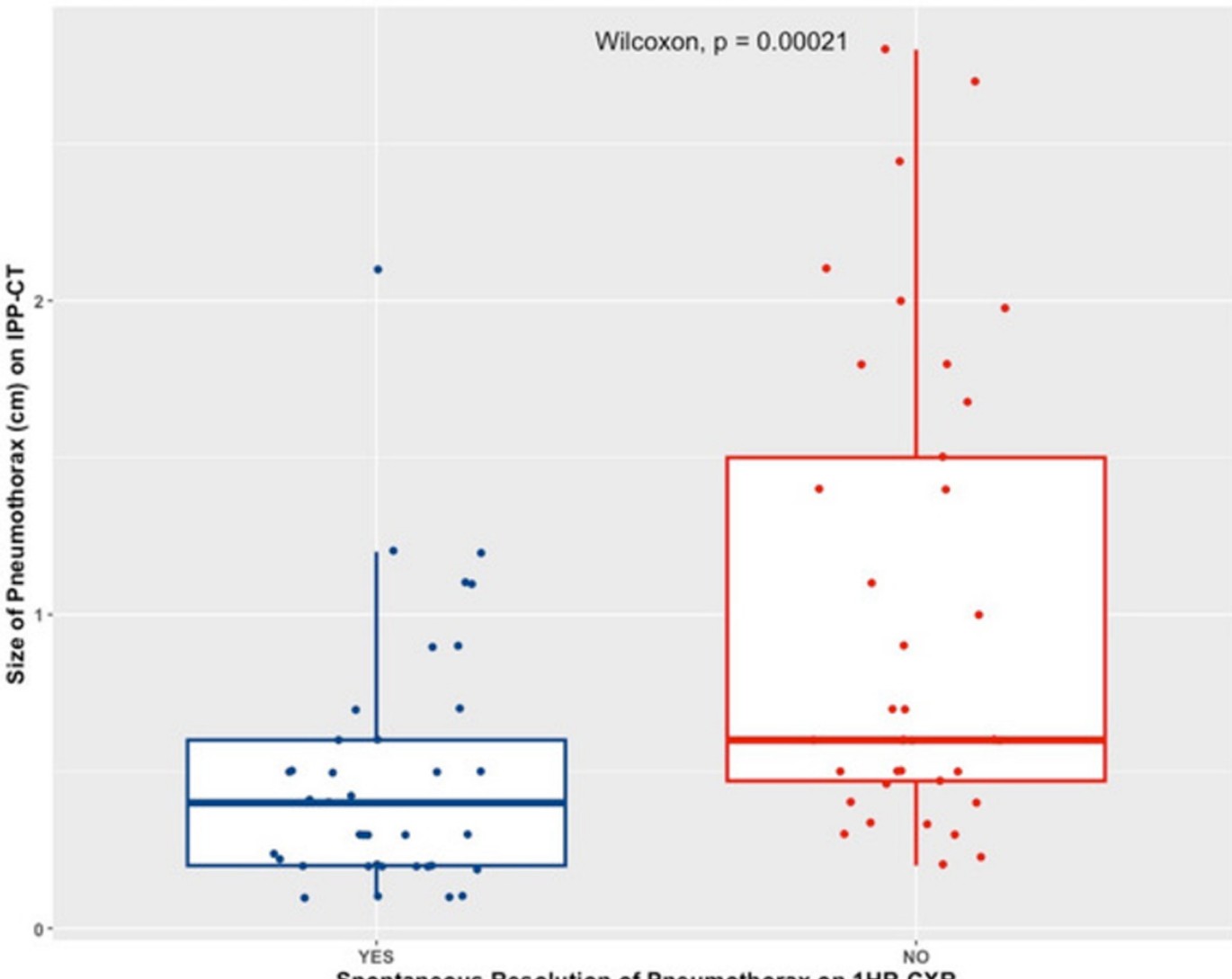

**Fig 2. Boxplot of pneumothorax size on immediate post-procedure CT.** The size of pneumothorax was significantly smaller in the group with spontaneous resolution of pneumothorax on 1HR-CXR than that in the group without spontaneous resolution.

### Other complications

Besides pneumothorax, two cases of hemoptysis (0.7%) developed as a result of biopsy. The two patients with hemoptysis were intubated for airway protection, admitted to the hospital for further management until complete resolution of hemoptysis, and eventually discharged within a week without additional procedural intervention. In the group without pneumothorax on IPP-CT or 1HR-CXR, no patient necessitated further imaging follow-up or returned to the hospital for symptoms related to delayed pneumothorax.

**Table 2. Comparison of associated variables between groups with and without pneumothorax.**

| | | No pneumothorax (n = 190) | Pneumothorax (n = 85) | p-value[b] |
|---|---|---|---|---|
| Demographics | Age | 63.3 ± 14.3 (18–91) | 64 ± 13.9 (25–88) | 0.69 |
| | Female | 87/190 (45.8%) | 41/85 (48.2%) | 0.81 |
| | Male | 103/190 (54.2%) | 44/85 (51.8%) | |
| COPD[a] | | 22/190 (11.6%) | 13/85 (15.3%) | 0.51 |
| Lesion size (cm) | | 2.5 ± 2 (0.6–16.3) | 2.4 ± 2.6 (0.6–15.7) | 0.88 |
| Biopsy device diameter | 20-gauge | 168/190 (88.4%) | 71/85 (83.5%) | 0.36 |
| | 18-gauge | 22/190 (11.6%) | 14/85 (16.5%) | |
| Biopsy device type | Full-core cutting | 123/190 (64.7%) | 54/85 (63.5%) | 0.33 |
| | Notched tip | 67/190 (35.3%) | 31/85 (36.5%) | |
| Needle tract embolization | Blood patch | 160/190 (84.2%) | 67/85 (78.8%) | 0.36 |
| | Gelfoam slurry | 30/190 (15.8%) | 18/85 (21.2%) | |
| Needle access | Anterior | 35/190 (18.4%) | 25/85 (29.4%) | 0.07 |
| | Posterior | 93/190 (48.9%) | 41/85 (48.2%) | |
| | Lateral | 62/190 (32.7%) | 19/85 (22.4%) | |
| Needle tract distance (cm) | | 1.7 ± 1.7 (0–7.8) | 2 ± 2 (0–9.2) | 0.18 |
| Needle access angle (°) | | 62.7 ± 19.5 (16–90) | 61.3 ± 20 (20–90) | 0.57 |
| Biopsy samples | Number | 4.3 ± 2 (2–17) | 4.7 ± 2.1 (2–13) | 0.16 |
| | ≤ 4 | 132/190 (69.5%) | 45/85 (52.9%) | 0.01 |
| | > 4 | 58/190 (30.5%) | 40/85 (47.1%) | |

The only statistically significant predictor of post-biopsy pneumothorax was biopsy sample number >4.

[a]COPD = chronic obstructive pulmonary disease

[b]p-values for continuous variables were obtained using student t-test or Mann-Whitney U test, and p-values for categorical variables were obtained using Pearson's Chi-square test or Fisher exact test

## Discussion

In our study, the complication rate of CT-guided lung biopsy was 31.6%, including pneumothorax (30.9%) and hemoptysis (0.7%). A chest tube was placed in 12.9% of all patients with pneumothorax and in 4% of all biopsy cases. These results are comparable to earlier reports, which have historically varied widely [4, 6–7, 12, 13]. In a systematic review and meta-analysis, the overall complication rate for core lung biopsy was 38.8%, with pooled pneumothorax and intervention rates of 25.3% (7–53.3%) and 5.6% (0–18.6%), respectively [4]. The Society of Interventional Radiology and the American College of Radiology also reported similar pneumothorax (12–45%) and chest tube placement rates (2–15%) [14]. Other potential complications associated with lung biopsy include pulmonary hemorrhage, hemoptysis, and hemothorax, with rates of 18%, 4.1%, and 0.4%, respectively [4, 6]. Although rare, air embolism and tumor seeding have also been reported [1, 4]. In our study, pneumothorax and hemoptysis were the only observed complications.

It has been reported that most cases of pneumothorax occur during or immediately after a biopsy (~91%), and a small number of pneumothoraces (2–9%) are usually detected only on follow-up chest radiography [6, 8, 15]. Pneumothorax identified only on a follow-up chest radiograph (i.e., not seen on IPP-CT) is referred to as a delayed pneumothorax in our study. The definition, incidence, management, and follow-up strategies for delayed pneumothorax are varied as no consensus or guidelines are available, and institutions follow individual protocols [6–8, 16]. In a study evaluating the incidence and risk factors of delayed pneumothorax, Choi et al. defined pneumothorax that had not developed up to 3 hours after procedure but

was detected later as a delayed pneumothorax, with an incidence rate of 3.3% [8]. They placed chest tubes in 20% of cases of delayed pneumothorax for symptoms and increasing size. Taleb et al. emphasized the importance of routine follow-up chest radiography as delayed pneumothorax was detected in 8.6% of all 278 biopsies, and 41.7% of those became clinically significant and necessitated chest tube placement [7]. Their study obtained the routine follow-up chest radiograph in 3.1 ± 2.9 hours after the biopsy. Brzezinski et al. reported rates of pneumothorax detected on IPP-CT and 2-hour chest radiography as 80.5% and 6%, respectively [17]. On the other hand, in a study of 336 CT-guided lung biopsies investigating how to safely shorten the observation time after biopsy, Ah-Lan et al. found obtaining a routine post-procedural chest radiograph unnecessary for patients with no symptoms and no pneumothorax on IPP-CT [6]. They did not obtain routine chest radiographs for 92.9% of asymptomatic patients and found only 2.2% of those developed delayed pneumothorax 2 to 10 days after the procedure. For 7.1% of asymptomatic patients who underwent 4-hour chest radiography, no pneumothorax was detected except for one patient with a delayed pneumothorax 7 days after the procedure. The authors concluded that routine serial post-procedural chest radiographs for asymptomatic patients would not help identify a delayed pneumothorax and also would not change management, as those patients without symptoms are less likely to require a chest tube [6]. Similarly, in our study, most pneumothorax was detected on IPP-CT (89.4%), only slightly more than half of pneumothoraces persisted on 1HR-CXR, and only a small number of that group required chest tube placement. A delayed pneumothorax was identified in only 3.3% (9/275) of all biopsies and 4.5% (9/199) of those who did not show pneumothorax on IPP-CT. None of those with delayed pneumothorax developed symptoms necessitating chest tube placement. Furthermore, after discharge, no patient developed a delayed pneumothorax requiring a return to the hospital or a hospital admission. Currently, our institution routinely performs 1HR-CXR after CT-guided lung biopsy and monitors the patient for an additional hour without further imaging before discharge.

Several factors are commonly associated with complications after percutaneous lung biopsy. The complication rates and risk factors differ between core biopsy and fine-needle aspiration (FNA) [4]. Also, the risk factors for pneumothorax seem to differ depending on the onset time. In a study evaluating risk factors of pneumothorax after core biopsy using a cutting needle, Lim et al. reported emphysema, smaller target size, deeper target location, and longer puncture time as risk factors associated with early pneumothorax [16]. In a similar study, Taleb et al. found that patients with delayed pneumothorax tended to have smaller lesion sizes, longer intrapulmonary needle tracts, and lower FEV1/FVC ratios than those without delayed pneumothorax [7]. Among those variables, the longer intrapulmonary tract was the only independent risk factor for delayed pneumothorax [7]. However, in a systematic review and meta-analysis including 32 core biopsy studies (8133 procedures) and 17 FNA studies (4620 procedures), no significant risk factor was identified for core biopsy [4]. Interestingly, hemoptysis and bleeding surrounding the puncture area were reported to be protective factors against the development of pneumothorax [16, 17]. Our study only included core biopsies using a device with either full-core cutting or notched needle design. We found that the intrapulmonary needle tract distance was the only significant risk factor for pneumothorax, and a smaller number of core samples was a protective factor against the development of pneumothorax. Due to a small number of hemoptysis cases in our study, we could not evaluate its role in protecting from pneumothorax.

Traditionally, an autologous blood patch has been the most common primary method for intrapulmonary needle tract embolization. In a prospective randomized study in 2013, intra-parenchymal autologous blood patch injection after percutaneous lung biopsy significantly reduced the rate of pneumothorax requiring chest tube placement (p = 0.048) [18]. This was

confirmed in another prospective randomized trial in 2019 which demonstrated that autologous blood patch injection was not inferior to a hydrogel plug regarding the rate of pneumothorax after percutaneous lung biopsy [19]. Recently, there has been an attempt to use Gelfoam slurry as an alternative method for embolizing the needle tract. Sum et al. conducted a retrospective study comparing the groups with and without tract embolization with Gelfoam slurry in patients undergoing a lung biopsy [20]. In their study, a group with Gelfoam slurry had a significantly lower rate of immediate pneumothorax (p = 0.032), and patients with emphysema were 2.4 times more likely to develop a delayed pneumothorax without Gelfoam. In our study, depending on operator preference, either autologous blood patch or Gelfoam slurry was randomly used for embolizing the intrapulmonary needle tract. When comparing the rates of immediate and delayed pneumothorax between the groups with blood patches and Gelfoam slurry, we found no significant differences. Although this was not our primary study endpoint, the result proved that Gelfoam is not inferior to autologous blood patch regarding the rate of pneumothorax.

Our study has several limitations. First, this is a single-center retrospective study with a relatively small number of subjects compared to previous similar studies. Second, we did not include any FNA as we rarely perform this type of procedure and therefore could not compare outcomes between core biopsy and FNA. Third, due to the retrospective nature of this study relying on the review of medical records, it is possible that some relevant information may have been lost regarding procedure details and patient follow-ups. Fourth, although all the patients were monitored for an additional hour following 1HR-CXR, the exact observation time thereafter may have varied depending on the recovery status of each patient after moderate sedation. Fifth, the criteria for needle aspiration and overall management of a small pneumothorax during or immediately after the procedure may also have varied depending on the operator.

In conclusion, our study results demonstrate that after a CT-guided percutaneous core lung biopsy, routine serial follow-up chest radiography can be limited to those who showed pneumothorax on IPP-CT. For those without pneumothorax on IPP-CT, a 1HR-CXR may only be necessary if the patient develops symptoms associated with potential pneumothorax.

## Supporting information

**S1 Dataset. Raw captured data.**
(CSV)

**S1 Table. Descriptive statistics of continuous variables.**
(TIF)

## Author Contributions

**Conceptualization:** Clayton Commander, Hyeon Yu.

**Data curation:** Lourens du Pisanie, Smith Ngeve, Clayton Commander, Hyeon Yu.

**Formal analysis:** Clayton Commander, Hyeon Yu.

**Investigation:** Clayton Commander, Hyeon Yu.

**Methodology:** Clayton Commander, Hyeon Yu.

**Project administration:** Clayton Commander, Hyeon Yu.

**Resources:** Clayton Commander, Hyeon Yu.

**Software:** Hyeon Yu.

**Supervision:** Clayton Commander, Hyeon Yu.

**Validation:** Clayton Commander, Hyeon Yu.

**Visualization:** Hyeon Yu.

**Writing – original draft:** Jared Thomas Weinand.

**Writing – review & editing:** Jared Thomas Weinand, Lourens du Pisanie, Clayton Commander, Hyeon Yu.

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
