## [Decision Letter · Decision Letter 0]

14 Feb 2023

PONE-D-22-29210Pneumothorax after computed tomography-guided lung biopsy: utility of immediate post-procedure computed tomography and one-hour delayed chest radiography

PLOS ONE

Dear Dr. Weinand,

Thank you for submitting your manuscript to PLOS ONE. After careful consideration, we feel that it has merit but does not fully meet PLOS ONE’s publication criteria as it currently stands. Therefore, we invite you to submit a revised version of the manuscript that addresses the points raised during the review process.

 I would like you to also share the data and  statistical analysis of your research in supplementary file section.

We look forward to receiving your revised manuscript.

Kind regards,

Muhammad Imran, MBBS, FCPS

Academic Editor

PLOS ONE

Journal Requirements:

2. Please note that in order to use the direct billing option the corresponding author must be affiliated with the chosen institute. Please either amend your manuscript to change the affiliation or corresponding author, or email us at plosone@plos.org with a request to remove this option.

Reviewers' comments:

Reviewer's Responses to Questions

**Comments to the Author**

1. Is the manuscript technically sound, and do the data support the conclusions?

Reviewer #1: Yes

Reviewer #2: Yes

2. Has the statistical analysis been performed appropriately and rigorously? 

Reviewer #1: Yes

Reviewer #2: Yes

3. Have the authors made all data underlying the findings in their manuscript fully available?

Reviewer #1: Yes

Reviewer #2: Yes

4. Is the manuscript presented in an intelligible fashion and written in standard English?

Reviewer #1: Yes

Reviewer #2: Yes

5. Review Comments to the Author

Reviewer #1: Manuscript is technically sound and the sample size is adequate. The language is simple and concise. Topic and study design is innovative. Data is presented in a simple and elaborative figures. Discussion is thorough and relevant, and has good comparison.

Reviewer #2: Authors of the article, "Pneumothorax after computed tomography-guided lung biopsy: utility of immediate post-procedure computed tomography and one-hour delayed chest radiography", have conducted a retrospective study to evaluate the utility of IPP-CT and 1Hr-CXR in diagnosing percutaneous lung biopsy induced pneumothorax. Furthermore, they have found needle tract distance and # of biopsy samples to be stastiscally significant risk factors associated with development of pneumothorax after percutaneous pulmonary biopsy. Although the study is retrospective in nature which may cause bias in conclusions. However, the strict exclusion criteria, logistic regression analysis and adequate number of sample size have strengthened the conclusions. The study has a valid rationale, study methodology, and comprehensive discussion of background literature supporting the conclusions drawn. I would like to encourage the authors to also share statistical analysis of their research in supplementary file section.

6. PLOS authors have the option to publish the peer review history of their article (what does this mean?). If published, this will include your full peer review and any attached files.

Reviewer #1: **Yes: **Munir Ahmad Khan

Reviewer #2: **Yes: **Hamza Yunus

---

## [Author Response · Author response to Decision Letter 0]

21 Feb 2023

1. Manuscript has been revised to comply with style requirements (including complete reworking of references section to comply with journal requirements/ICMJE guidance), and file naming for this submission has been observed to comply with journal requirements.

2. An email has been sent requesting cancellation of the previous direct institutional billing request.

3. A dedicated ethics statement has been added to the materials and methods section, and per reviewer request, additional institutional IRB details have been added.

4. As noted above, reference list has been reformatted to comply with journal requirements.

---

## [Decision Letter · Decision Letter 1]

27 Mar 2023

Pneumothorax after computed tomography-guided lung biopsy: utility of immediate post-procedure computed tomography and one-hour delayed chest radiography

PONE-D-22-29210R1

Dear Dr, Weinand,

We’re pleased to inform you that your manuscript has been judged scientifically suitable for publication and will be formally accepted for publication once it meets all outstanding technical requirements.

Kind regards,

Muhammad Imran, MBBS, FCPS

Academic Editor

PLOS ONE

Additional Editor Comments (optional):

Reviewers' comments:

Reviewer's Responses to Questions

**Comments to the Author**

1. If the authors have adequately addressed your comments raised in a previous round of review and you feel that this manuscript is now acceptable for publication, you may indicate that here to bypass the “Comments to the Author” section, enter your conflict of interest statement in the “Confidential to Editor” section, and submit your "Accept" recommendation.

Reviewer #2: All comments have been addressed

2. Is the manuscript technically sound, and do the data support the conclusions?

Reviewer #2: Yes

3. Has the statistical analysis been performed appropriately and rigorously? 

Reviewer #2: Yes

4. Have the authors made all data underlying the findings in their manuscript fully available?

Reviewer #2: Yes

5. Is the manuscript presented in an intelligible fashion and written in standard English?

Reviewer #2: Yes

6. Review Comments to the Author

Reviewer #2: The revised manuscript addresses the concerns raised by the reviewers. The article has clarity and is technically sound to be considered for publication.

7. PLOS authors have the option to publish the peer review history of their article (what does this mean?). If published, this will include your full peer review and any attached files.

Reviewer #2: **Yes: **Hamza Yunus

---

## [Editor Report · Acceptance letter]

10 Apr 2023

PONE-D-22-29210R1 

­Pneumothorax after computed tomography-guided lung biopsy: utility of immediate post-procedure computed tomography and one-hour delayed chest radiography 

Dear Dr. Weinand:

I'm pleased to inform you that your manuscript has been deemed suitable for publication in PLOS ONE. Congratulations! Your manuscript is now with our production department. 

Kind regards, 

on behalf of

Dr. Muhammad Imran 

Academic Editor

PLOS ONE